# Description of Two New Cases of *AQP1* Related Pulmonary Arterial Hypertension and Review of the Literature

**DOI:** 10.3390/genes13050927

**Published:** 2022-05-22

**Authors:** Natalia Gallego-Zazo, Alejandro Cruz-Utrilla, María Jesús del Cerro, Nuria Ochoa Parra, Julián Nevado Blanco, Pedro Arias, Pablo Lapunzina, Pilar Escribano-Subias, Jair Tenorio-Castaño

**Affiliations:** 1Institute of Medical and Molecular Genetics Molecular (INGEMM), Hospital Universitario La Paz Research Institute (IdiPaz), Hospital Universitario La Paz, 28046 Madrid, Spain; natalia.gallego.zazo@idipaz.es (N.G.-Z.); jnevadobl@gmail.com (J.N.B.); palajara@gmail.com (P.A.); plapunzina@gmail.com (P.L.); 2CIBERER, Center for Rare Disease Network Research, Instituto de Salud Carlos III, 28029 Madrid, Spain; 3ITHACA, European Reference Network on Rare Congenital Malformations and Rare Intellectual Disability, 75019 Paris, France; 4Pulmonary Hypertension Unit, Department of Cardiology, Hospital Universitario 12 de Octubre, 28041 Madrid, Spain; acruzutrilla@gmail.com (A.C.-U.); nuriaochoaparra@hotmail.com (N.O.P.); pilar.escribano.subias@gmail.com (P.E.-S.); 5Center for Networked Biomedical Research in Cardiovascular Diseases, Instituto de Salud Carlos III (CIBERCV), 28029 Madrid, Spain; 6Pulmonary Hypertension Unit, Hospital Universitario Ramón y Cajal, 28034 Madrid, Spain; majecerro@yahoo.es; 7ERN, European Reference Network Pulmonary Hypertension, 60590 Frankfurt, Germany

**Keywords:** *AQP1*, pulmonary arterial hypertension, massive paralleled sequencing, NGS, genomic medicine, personalized medicine

## Abstract

Pulmonary arterial hypertension (PAH) is a severe clinical condition characterized by an increase in mean pulmonary artery pressure, which leads to a right ventricular hypertrophy and potentially heart failure and death. In the last several years, many genes have been associated with PAH, particularly in idiopathic and heritable forms but also in associated forms. Here we described the identification of two unrelated families in which the *AQP1* variant was found from a cohort of 300 patients. The variants were identified by whole exome sequencing (WES). In the first family, the variant was detected in three affected members from a hereditary PAH, and in the second family the proband had PAH associated with scleroderma. In addition, we have reviewed all cases published in the literature thus far of patients with PAH and *AQP1* variants. Functional studies have led to some contradictory conclusions, and the evidence of the relationship of *AQP1* and PAH is still limited. However, we describe two further families with PAH and variants in *AQP1*, expanding both the number of cases and the clinically associated phenotype. We provide further evidence of the association of *AQP1* and the development of hereditary and associated forms of PAH.

## 1. Introduction

PAH is a progressive cardiopulmonary disease characterized by a persistent elevated mean pulmonary artery pressure (mPAP ≥ 20 mmHg) with a mean pulmonary arterial wedge pressure (PAWP) ≤ 15 mmHg and pulmonary vascular resistance (PVR) ≥ 3 UW [1]. These features can lead to right heart dysfunction, right heart failure, and ultimately death if untreated [2,3,4]. According to the latest classification, PAH in adults includes several clinical subgroups with almost identical obstructive pathologic changes in the pulmonary arteries: idiopathic PAH (IPAH), hereditary PAH (HPAH), drug/toxin induced, connective tissue disease (CTD), HIV infection, portal hypertension, congenital heart disease (CHD), schistosomiasis, pulmonary venooclusive disease (PVOD) or pulmonary capillary hemangiomatosis (PCH), and persistent pulmonary hypertension of the newborn (PPHN) [1]. PAH has a variable etiology and clinical expressivity, making the clinical diagnosis an occasional challenge.

Current clinical classification of pulmonary hypertension (PH) included five groups based on clinical and etiological criteria: Group 1 or pulmonary arterial hypertension (PAH); Group 2 or PH due to left heart disease; Group 3 or PH due to lung diseases and/or hypoxia; Group 4 or PAH due to pulmonary artery obstruction; Group 5 or PH with unclear multifactorial mechanisms [1].

The occlusive arteriopathy occurs because of chronic obstruction of small pulmonary arteries due to alterations in the structure and function of the endothelium and vascular smooth muscle cells [5]. All these changes are driven by a combination of vasoconstriction, thrombosis, inflammation, and proliferative and obstructive remodelling of the pulmonary artery wall through a growth of neointimal, medial, and adventitial layers [6]. The precise process implicated in the development of the disease is still unclear, however, vascular remodelling in PAH is a complex and multifactorial process and it is hypothesized that it might be an interaction between genetic predisposition and environmental risk factors [5]. Therefore, more information is needed to evaluate the possible involvement of additional factors in its pathogenesis.

The first evidence of the genetic contribution was observed in 2000 when pathogenic variants in bone morphogenetic protein receptor type 2 (*BMPR2*), which codifies a protein member of the transforming growth factor-β (TGF-β) family, were associated with the disease [7,8]. However, variants in this gene are detected in 20% of patients with IPAH and about 60% of HPAH individuals. Moreover, *BMPR2* has an incomplete penetrance of 20% [9]. In addition, there is a certain number of patients in whom a genetic diagnosis is not achieved.

Advances in massive paralleled sequencing in PAH have allowed the identification of several new causative and susceptibility genes associated with the disease. In 2018, a research study performed on a European cohort of over 1000 adult-onset patients with PAH confirmed the presence of potentially causative variants in approximately 19% of the patients in previously well-known associated genes, including *BMPR2*, *TBX4*, *ACVRL1*, *ENG*, *SAMD9,* and *KCNK3* [10]. Additionally, they found variants in other genes in approximately 4% of the patients. These results suggest that there can be still unknown additional genetic factors that contribute to disease development [10]. One of those proposed genes was *AQP1*. *AQP1* encodes for aquoporin-1 (AQP1), a member of a family of membrane channel proteins that increases cell membrane water permeability [11]. It is expressed widely in the vascular endothelia, and it promotes endothelial cell migration and angiogenesis [12]. To date, 11 variants in *AQP1* have been described in 17 PAH patients diagnosed with IPAH and HPAH [10,13].

Several genes are currently related to PAH with a variable degree of evidence. According to the pulmonary hypertension gene curation expert panel of pulmonary hypertension of ClinGen [14], the association of *BMPR2, KCNK3, KDR, SMAD9,* and *TBX4* and PH is definitive, *GDF2* is strong, and *AQP1* has a limited disease relationship.

Therefore, in order to address the molecular diagnosis of PAH patients, we initially performed massive paralleled sequencing through a customized panel which included 21 genes (HAP v1.2) [15]. However, about 68% of the tested patients did not indicate any pathogenic or likely pathogenic variant in these genes; thus, we have performed whole exome sequencing (WES) in non-conclusive patients evaluated with the custom panel. Here, we present two PAH unrelated families in which we have identified a missense variant in *AQP1* [10] in patients with HPAH and PAH-CTD. In addition, we have reviewed all previously described cases with *AQP1* variants published to date. Our results add additional evidence of the association of *AQP1* variants with the development of PAH, not only with idiopathic and hereditary forms, but also in an associated form, such as scleroderma.

## 2. Materials and Methods

Since November 2011, our institute offers genetic studies for patients with presumptive diagnosis of PAH, mainly belonging to the adults and child Spanish PAH registries, REHAP (Spanish Registry of Pulmonary Arterial Hypertension) (https://www.rehap.org/) and REHIPED (Spanish Registry of Pediatric with Pulmonary Hypertension) (https://www.rehiped.org/). Both registries include PAH patients with different etiologies. All patients have been informed and participants or legal tutors were invited to fill out an informed consent agreement before their inclusion in the project. This project was approved by the ethical committee for research at La Paz University Hospital (CEIC-HUL PI-1210). For all included patients, DNA was extracted from peripheral blood according to standard procedures.

### 2.1. Case Presentation

Here we present the results of genetic analysis in two unrelated families.

#### 2.1.1. Family 1

The index patient of Family 1 is a female, diagnosed with IPAH at the age of 7 years. Currently she is 13 years old and she receives double oral therapy with iPDE5 and ERA. During the diagnostic, right heart catheterization (RHC), mean pulmonary artery pressure (mPAP) was 42 mmHg, pulmonary artery wedge pressure (PAWP) was 13 mmHg, and pulmonary vascular resistance (PVR) was 10.6 wood units (WU). The father of the proband was also affected and diagnosed after the index case; so, PAH subtypes were revised to HPAH. Although he suffered from syncopes of unexplained cause in 2007, he was later diagnosed with PAH in 2011 (at 38 years old). RHC indicated that mPAP was 45 mmHg, PAWP was 15 mmHg, and PVR was 3.7 WU. The mother and sister (22 years) of the index patient have also been studied but neither presented symptoms suggestive of the disease and there was no known relevant family history until very recently. The older sister of the proband was diagnosed during the preparation of this article and she has indicated very mild PAH according to the results of the catheterization (Table 1).

#### 2.1.2. Family 2

The index case of Family 2 is a female with a PVOD-like PAH involvement associated with limited systemic sclerosis (PAH-SSc). She was diagnosed when she was 62 years old due to a progressive impairment of her functional capacity, and severe respiratory insufficiency. The initial RHC had a mPAP of 41 mmHg, PAWP was 3 mmHg, and PVR was 16.5 WU. Interestingly, the patient indicated a severe impairment of the diffusing capacity of the lung for carbon monoxide (DLCO of 23%) and thoracic computed tomography indicated lymph node enlargement and septal lines, two of the three radiological findings described as typical of the PVOD phenotype [16]. Unfortunately, despite double sequential therapy with intravenous epoprostenol and endothelin receptor antagonist, the patient died 2 years after diagnosis due to sudden cardiac death.

### 2.2. Genetic Analysis

A customized NGS panel of 21 genes (HAP v1.2) was designed in-house and included: *ABCC8*, *ACVRL1*, *BMPR1B*, *BMPR2*, *CAV1*, *CBLN2*, *CPS1*, *EIF2AK4*, *ENG*, *GDF2*, *KCNA5*, *KCNK3*, *MMACHC*, *NOTCH3*, *SARS2*, *SMAD1*, *SMAD4*, *SMAD5*, *SMAD9*, *TBX4*, and *TOPBP1* [15]. All families presented in this article were nonconclusive for this panel, so we extended the genetic analysis by applying a WES. We performed WES in patients and available relatives. Library preparation was carried out using the Agilent SureSelect ^M^ (v 6.0) all-exon kit followed by sequencing in a NovaSeq6000 Sequencer (Illumina, San Diego, CA, USA). The exomes were analyzed by VarSeq (Golden Helix, Bozeman, MT, USA) to detect both single nucleotide variants (SNVs) and copy number variants (CNVs). We have developed an in-house prioritization algorithm for WES analysis (Figure 1).

Prioritization algorithm included the application of several step-by-step custom filters. The first step was to filter out variants which had an insufficient quality: variants with less than 20 reads, less than 90% of genotype quality, and variants with less than 20% of variant allele frequency. Second, only variants which had an allele frequency below 1% (≤0.01) compared to all the following pseudocontrol population databases: gnomAD exomes (v3.1), gnomAD genomes (v3.0), Kaviar (version 160204-Public), or Bravo (TOPMed Freeze 8) were kept. Then, we filtered based on the pathogenicity, assessed by the analysis of several bioinformatic tools included in the dbNSFP (v4.0) [17] database plus the computation of the CADD score (v1.6) [18]. Finally, variants were classified according to the ACMG (American College of Medical Genetics) guidelines [19].

Additionally, in order to carry out the CNVs analysis, we applied a custom script developed in-house called “LACONv” v1 (https://github.com/kibanez/LACONv, accessed on 2 December 2021).

## 3. Results

Two missense variants in *AQP1* (Table 2) were identified in two unrelated families.

The pedigree of Family 1 is displayed in Figure 2A. The index patient (HTP337) indicated a heterozygous missense variant in *AQP1:* NM_198098.3:c.376C>T:p. (Arg126Cys) (Table 2). This variant has been inherited from their affected father by both the index case and her affected sister (HTP336). The mother was also studied and she did not present any symptom of PAH or the heterozygous variant.

In Family 2, we detected a heterozygous missense variant *AQP1:* NM_198098.3:c.423C > G:p. (Ile141Met) (Table 2) in a patient diagnosed with PAH-SSc. In this case, it was not possible to perform the segregation analysis of the variant because it was not possible to obtain DNA from the parents.

Both variants were absent in several pseudocontrol population databases (Table 2). In silico analysis suggested a deleterious effect of the two missense variants and, according to ACMG (American College of Medical Genetics) guidelines, these variants were classified as variants of unknown significance (VUS).

Conversely, the CNVs analysis did not reveal any genomic rearrangement.

## 4. Discussion

PAH is an infrequent disease, with low incidence in the general population but with a devastating prognosis without proper treatment [20,21]. Despite the efforts made by the scientific community in the study of the disease, there are only a few well-known pathways associated to this pathology and that could be used to settle a treatment for patients [22]. Given the known role of genetics in the development of the disease and the advance in massive parallel sequencing in PAH, several new causative and susceptibility genes were associated with the pathogenesis of PAH.

Therefore, to improve the overall diagnosis in our country, we have performed the analysis through WES of our patients and all the available relatives in order to establish a genetic diagnosis, to analyze segregation of the variants detected, and to study the genotype–phenotype correlation.

For this purpose, we have designed a custom filtering algorithm that has allowed us to detect two heterozygous missense variants in *AQP1* in two unrelated PAH patients (Table 2) from two different families. Pathogenic variants in *AQP1* were associated with PAH for the first time by Gräf et al. in 2018. In their study, a cohort of 1048 PAH patients were analyzed by whole genome sequencing (WGS), revealing a higher frequency of rare variants in *AQP1* in PAH patients compared to matched controls [10].

Patient 1 from Family 1 is a female individual with hereditary PAH (HAPH) who was diagnosed at 7 years of age. After WES and variant prioritization we have identified a heterozygous missense variant in *AQP1:* NM_198098.3:c.376C > T:p. (Arg126Cys). Analysis of her relatives confirmed that the variant was inherited from her father, who is also affected (Figure 2A). In this family, the father was diagnosed with PAH in 2011 at the age of 38 years, approximately 2 years after first presenting with clinical symptoms. Currently, he remains stable under double therapy with systemic epoprostenol and sildenafil. In both cases (father and daughter), the clinical course is benign with hemodynamic severity and without PVOD characteristics. Interestingly, the older sister of the proband remained apparently healthy until shortly before the publication of this article. Fortunately, this patient has been diagnosed early. However, the description of this family appears to support the hypothesis of genetic anticipation, which causes a younger age of onset in subsequent generations. In this sense, we emphasize the importance of an early genetic diagnosis of the disease in order to ensure an adequate follow-up, both for patients and their relatives, and correct genetic counseling for the families. Noteworthy, the father and the index case presented severe PAH, while the sister of the proband had very mild symptoms, suggesting a variable expressivity that leads to a different severity of the disease.

The variant present in this patient (NM_198098.3:c.376C > T) has been previously reported in two studies [10] but unfortunately, we do not have information on the segregation in the other families. However, the variant is absent from several population databases (Table 2). Given the fact that this variant was previously described in three additional unrelated cases, we suggested a probable hotspot that may be associated with the development of PAH. Thus, this variant appears to be a hotspot in PAH as it has been detected in several independent studies [10,13]. The variant is not located within a specific protein domain, although it can alter the structural conformation of the protein between the two transmembrane domains.

Patient 2 from Family 2 is a 62-year-old woman who was diagnosed with PAH associated with scleroderma (SSc) and PVOD-like lesions. She suffered a sudden cardiac death 2 years after the diagnosis. In addition, due to her age at diagnosis, segregation of the variant in her parents was not possible (Figure 2B).

The identification of the *AQP1* variant in PAH-SSc suggests that the presence of pathogenic variants in *AQP1* may be involved not only in the development of primary forms of PAH, such as idiopathic or hereditary forms, but also in other associated forms such as PAH associated with scleroderma. The occurrence of PH in CTD patients has a major impact on their quality of life and prognosis [23]. Additionally, a higher venous involvement has been described in SSc-related PAH, which could be associated with a more challenging management and a poorer survival rate [24].

Therefore, identification of pathogenic variants associated with these infrequent forms can help to assess an early diagnosis and proper genetic counseling, which is crucial to improve survival in these individuals and to perform a correct follow-up. In fact, the association between gene expression and severity of PAH in PAH-SSc patients has been previously demonstrated several times. The results obtained by Grigoryev et al. suggested an association between angiogenesis-related gene expression and severity of PAH in PAH-SSc patients and provided evidence that gene expression is different depending on the level of severity [25]. Additionally, Zhen et al. indicated that some genes may serve as potential biomarkers in SSc-PAH [26]. To the best of our knowledge, this is the first time that it is described as a pathogenic variant in *AQP1* in a patient with PAH associated with scleroderma and severe impairment of diffusing capacity of the lung for carbon monoxide.

To date, 11 different variants have been published in two previous studies [10,13] (Summarized in Table 2). The available clinical information from patients (Table 1) indicates that the mean age of diagnosis of patients reported by Gräf at al. is 32.2 years. In our study, we describe a pediatric patient and his father, diagnosed at 38 years of age. Therefore, the late diagnosis at the age of 62 years of the HTP373 patient should be noted. Likewise, this patient has a marked ventilatory insufficiency not observed in any previous patient. Altogether, these data suggest that clinical phenotype and age of onset of PAH due to pathogenic variants in *AQP1* is highly heterogeneous and variable.

Variants in AQP1 described thus far are located across all exons and affect all protein domains (Figure 3B), suggesting that there is no hot-spot region within the protein, except for the recurrent variant p.Arg126Cys detected in three unrelated cases.

The role of pathogenic variants in *AQP1* in the development of PAH is currently not fully elucidated. Gräf et al. suggested that AQP1 is predominantly localized in the pulmonary endothelium in the normal human lung. In addition, they determined that there is expression of *AQP1* in primary cultures of pulmonary artery smooth muscle cells (PASMCs) and in pulmonary endothelial cells (PAECs) [10].

Nevertheless, in 2017 and in 2019, two studies in cell cultures of the hypoxia-induced pulmonary hypertension mouse models determined that *AQP1* knockout reduced proliferation and migration potential, and increased proliferation in PASMCs and PAECs. *AQP1* expression is increased after hypoxia in PASMCs and PAECs in mice, and *AQP1* knockout attenuates hypoxic PH in mice, reduced RVP, and proliferation in vivo [27,28].

This study has several limitations. First, the technology used for the genetic analysis variants does not allow the identification of variants outside of exons or exon-intron boundaries, which may hold essential regulatory elements. Secondly, it was not possible to study the segregation of the variants in all cases and the study of the variants detected in this study was carried out according to in silico predictors. Therefore, it would be desirable to develop in vitro studies aimed at the functional characterization of the specific variants detected to confirm their roles in the pathogenesis of the disease. We also recommend adding this gene to all panels for the study of genes associated with PAH aiming to detect more patients.

## 5. Conclusions

The role of the variants in *AQP1* is currently controversial. Our results demonstrate familial segregation of the *AQP1* variant in the family of Patient 1, which provides further support for the potentially causative role of *AQP1* variants in primary and associated forms of PAH. Here, we describe a family in which we observed a clear co-segregation of the variant with the disease, a variant that has been previously detected in two other unrelated families with PH. In addition, we suggest a variable expressivity of the disease which leads to a different severity of PH. In acknowledgement of these results, we add more evidence that missense variants in *AQP1* can be implicated in the development of PAH. However, further studies, particularly functional validation in in vitro models, are required to confirm the functional impact of pathogenic variants in *AQP1* and its implication role in PAH.

## Figures and Tables

**Figure 1 genes-13-00927-f001:**
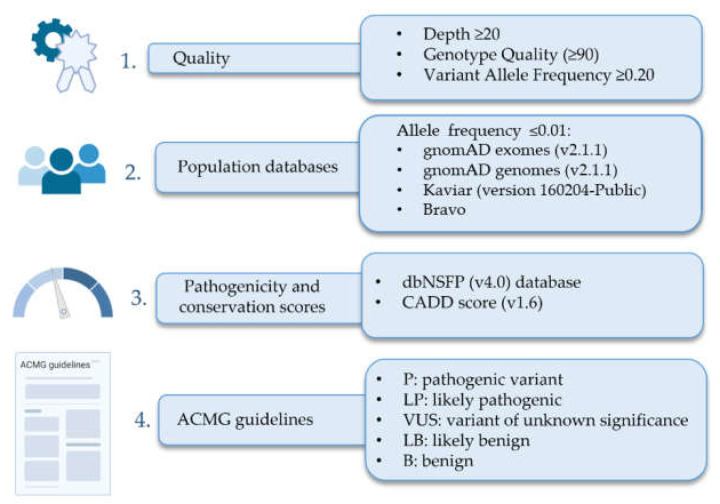
Filter algorithm designed to analyze SNVs variants detected by whole exome sequencing analysis.

**Figure 2 genes-13-00927-f002:**
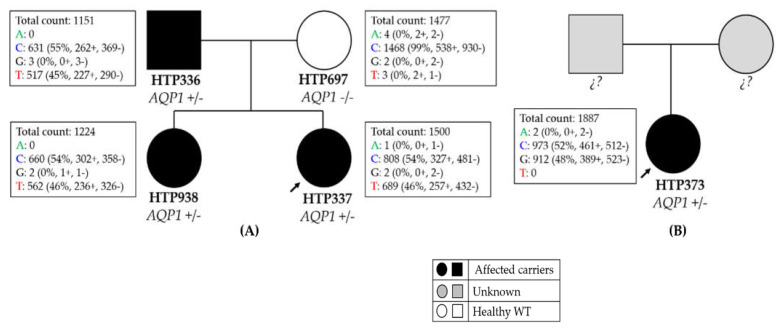
Pedigree of the families with *AQP1* variants. (**A**) Pedigree of Family 1. The heterozygous missense variant at nucleotide position c.376 (c.376C > T) in *AQP1* (NM_198098.3) causes an amino acid substitution of an arginine to cysteine at position 126 (p. Arg126Cys). The same heterozygous missense variant was found in her affected father and sister. (**B**) Pedigree of Family 2. The heterozygous missense variant at nucleotide position c.423 (c.423C > G) in *AQP1* (NM_198098.3) causes an amino acid substitution from isoleucine to methionine at position 141 (p. Ile141Met) of the protein. Samples of these parents were not available. Legend: +/+ homozygous for the alternative allele; +/− heterozygous. Boxes indicate the information of the reads obtained in the WES. Legend: + forward reads; − reverse reads.

**Figure 3 genes-13-00927-f003:**
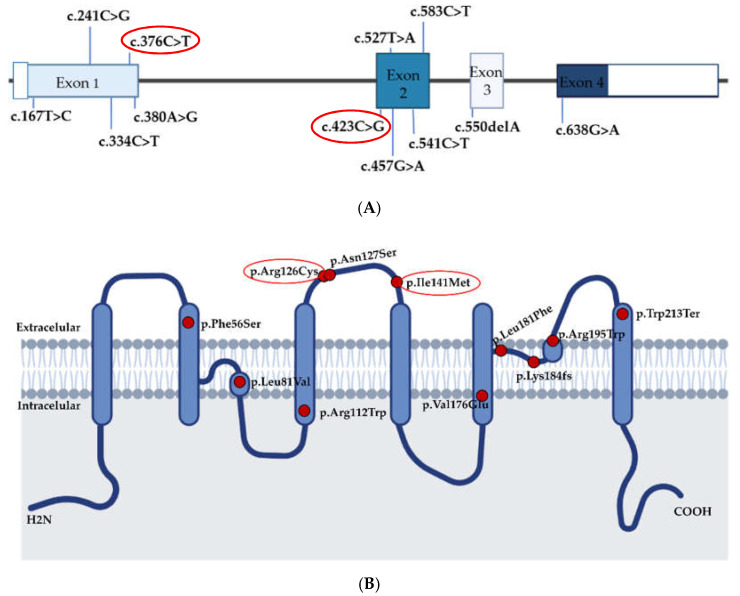
Variants described in *AQP1*. (**A**) Localization of cDNA variants detected in PAH cases in *AQP1*. (**B**) Representation of AQP1 protein and the amino acid changes detected in PAH cases. The two variants described in this study are labeled in the red circle; both are located at extracellular domains of the AQP1 protein. Variants were annotated with NM_198098.3.

**Table 1 genes-13-00927-t001:** Clinical information of PAH patients with *AQP1* variants. Clinical data from Gräf et al. and Wang et al. are estimated as the mean from the cases described [10,13].

		This Study			Gräf et al., 2018	Wang et al., 2019
Patient	Family 1 (HTP336)	Family 1 (HTP337)	Family 1 (HTP938)	Family 2 (HTP373)	N = 9	N = 8
Sex	Male	Female	Female	Female	4 females + 5 males	NA
PAH diagnosis	HPAH	HPAH	HPAH	PAH-SSc, PVOD-like	2 HAPH + 7 IPAH	8 IPAH
Age at diagnosis (years)	38	7	22	62	32.2 (25.3-46.2)	NA
Current age (years)	48	13	22	Sudden cardiac death 2 years after diagnosis	NA	NA
NYHA functional class	IV	II	I	III	IV (50%) III (37.5%) I (12.5%)	NA
MPAP (mmHg) PAWP (mmHg) PVR (Wood units) DLCO (%) CO (L/min)	45 15 5.2 120 5.8	42 13 10.6 NA NA	37 11 4.93 90 5.7	41 3 16.5 23 NA	61.5 (48.0-68.2) 8 (7.5-8.0) NA 81 (78.2-91.0) 4.7 (4.3-5.3)	NA NA NA NA NA
6MWT (meters)	480	516	NA	310	NA	NA
FVC FEV1 FEV1/FVC (%)	2.39 2.20 92	NA NA NA	NA NA NA	NA NA NA	NA NA NA	NA NA NA
Last known PAH therapies	Double initial therapy including intravenous epoprostenol and sildenafil	Initial double oral therapy with iPDE5 and ERA	NA	Double sequential therapy with intravenous epoprostenol and ERA	NA	NA

Clinical information of individuals with *AQP1* variants. HPAH: Hereditary pulmonary arterial hypertension; IPAH: idiopathic pulmonary arterial hypertension; PAH-SSc: pulmonary arterial hypertension associated with scleroderma; NYHA: New York Heart Association functional class; MPAP: Mean pulmonary arterial pressure; PAWP pulmonary arterial wedge pressure; PVR: pulmonary vascular resistance; DLCO: diffusing capacity of the lung for carbon monoxide; CO: cardiac output; 6MWT: 6 min walking test; ERA: endothelin receptor antagonists; NA: not available; iPDE5: inhibitors of phosphodiesterase 5, FVC: forced vital capacity, FEV1: forced expiratory volume in 1 second.

**Table 2 genes-13-00927-t002:** Information of variants in *AQP1* described in PAH cases.

Chr. Coordinate	PAH Patients with Variant	cDNA Position ^1^	Protein Position	Variant Effect	Population Frequency ^2^	DbSNFP ^3^ + CADD	ACMG ^5^	Reference
Chr7:30951691	1	c.167T > C	p.Phe56Ser	Missense	Absent	16/1 + 28	NA	[13]
Chr7:30961786	2	c.241C > G	p.Leu81Val	Missense	Absent	6/6 + 10	NA	[13]
Chr7:30962212	1	c.334C > T	p.Arg112Trp	Missense	Absent	16/1 + 35	NA	[13]
Chr7:30951900	5	c.376C > T	p.Arg126Cys	Missense	Absent	10/6 + 33	VUS ^6^	[10,13] This study
Chr7:30951904	1	c.380A > G	p.Asn127Ser	Missense	0.00000834	14/3 + 25	NA	[13]
Chr7:30961719	1	c.423C > G	p.Ile141Met	Missense	Absent	6/10 + 22	VUS	This study
Chr7:30961753	1	c.457G > A	p.Val153Met	Missense	Absent	15/1 + 27	NA	[13]
Chr7:30961823	2	c.527T>A	p.Val176Glu	Missense	Absent	17/0 + 33	NA	[10]
Chr7:30961837	1	c.541C > T	p.Leu181Phe	Missense	Absent	17/0 + 29	NA	[10]
Chr7:30963232	1	c.550delA	p.Lys184fsTer18	frameshift	Absent	NA ^4^	NA	[13]
Chr7:30962212	4	c.583C > T	p.Arg195Trp	Missense	0.0000319	18/0 + 35	NA	[10]
Chr7:30963072	1	c.638G > A	p.Trp213Ter	Nonsense	Absent	8/1 + 43	NA	[10]

^1^ Human reference genome is hg19. The transcript used for variant annotation was *AQP1:* NM_198098.3; ^2^ AF: Population frequency was obtained from gnomAD genomes; ^3^ Predictors that suggest deleterious effect/predictors that suggest benign effect; ^4^ NA: not available; ^5^ ACMG: American College of Medical Genetics and Genomics classification; ^6^ VUS: variants of unknown significance.

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
