# Peer review of "Description of Two New Cases of AQP1 Related Pulmonary Arterial Hypertension and Review of the Literature"

_genes, 2022, doi:10.3390/genes13050927_

Round 1

Reviewer 1 Report

The manuscript is well written and formulated nicely. Apart from some minor improvements, it is ready for readers.

The association between gene expression and severity of PAH in PAH-SSc patients has been demonstarted (Grigoryev et al. 2009, Ji-Na Zheng et al. 2020). It is advisable to mention it in your manuscript and ellaborate it.

Page 4, last line: it is not clear they used all those mentioned population databases or they used them randomly.

Page 5, line 188: please add a T: it was not.

Reviewer 2 Report

The manuscript ‘Description of two new cases of AQP1 related Pulmonary Arterial Hypertension and review of the literature’ is an interesting piece of work especially when studies in the past have led to contradictory conclusion. The role of AQP1 in the development of PAH has been discovered.

The abstract is long (exceeds 200 words). I would suggest the authors to shorten it below 200-word count.

Likewise, the introduction is too long. The number of words and paragraphs should be reduced.

The authors haven’t mentioned about the statistical analysis performed in this study. It should be included under a separate heading.

How did the authors calculate the sample size?

There is no conclusion section. Though, the last paragraph of discussion seems to be conclusion, nonetheless it should be written under a separate heading.

Owing to contradictory findings in the literature and in this study as well, the authors should mention ‘Limitations of the study’ and preferably also include ‘Future directions’.

Some references cited in the text are very old. It would be better if they are replaced with the newer ones.
